# Novel Insights into the Role of Keratinocytes-Expressed TRPV3 in the Skin

**DOI:** 10.3390/biom13030513

**Published:** 2023-03-10

**Authors:** Yaotao Guo, Yajuan Song, Wei Liu, Tong Wang, Xianjie Ma, Zhou Yu

**Affiliations:** 1Department of Plastic Surgery, Xijing Hospital, Fourth Military Medical University, Xi’an 710032, China; 2Department of the Cadet Team 6 of School of Basic Medicine, Fourth Military Medical University, Xi’an 710032, China

**Keywords:** keratinocytes, TRPV3, itch, heat pain, hair development, skin regeneration, olmsted syndrome

## Abstract

TRPV3 is a non-selective cation channel that is highly expressed in keratinocytes in the skin. Traditionally, keratinocytes-expressed TRPV3 is involved in multiple physiological and pathological functions of the skin, such as itching, heat pain, and hair development. Although the underlying mechanisms by which TRPV3 functions in vivo remain obscure, recent research studies suggest that several cytokines and EGFR signaling pathways may be involved. However, there have also been other studies with opposite results that question the role of TRPV3 in heat pain. In addition, an increasing number of studies have suggested a novel role of TRPV3 in promoting skin regeneration, indicating that TRPV3 may become a new potential target for regulating skin regeneration. This paper not only reviews the role of keratinocytes-expressed TRPV3 in the physiological and pathological processes of itching, heat pain, hair development, and skin regeneration, but also reviews the relationship between TRPV3 gene mutations and skin diseases such as atopic dermatitis (AD) and Olmsted syndrome (OS). This review will lay a foundation for further developing our understanding of the mechanisms by which TRPV3 is involved in itching, heat pain, and hair development, as well as the treatments for TRPV3-related skin diseases.

## 1. Introduction

Transient receptor potential (TRP) channels are a mass group of cation channels that are widely expressed on the plasma membrane of diverse cell types. TRP channels are tetramers formed by TRP proteins. Twenty-seven kinds of human TRP channels have been discovered [1]. TRP channels respond to a variety of extracellular and intracellular stimuli, such as temperature, pH, osmotic pressure, cytokines, exogenous chemicals, physical cues, etc. [2,3,4,5]. According to their amino acid sequence and topological structures, the TRP channel family is divided into 8 subfamilies, including TRPA (ankyrin), TRPC (canonical), TRPM (melastatin), TRPML (mucolipin), TRPN (Drosophila NOMPC), TRPP (polycystin), TRPV (vanilloid), and TRPY (yeast) (Figure 1) [1].

TRPV channels are a class of highly conserved TRP channels that consists of six members (TRPV1-6) and can be activated by vanillin, vanillic acid, and capsaicin [6]. Among the TRPV channels, TRPV1-4, also known as thermoTRPV channels, are thermosensitive non-selective cation channels, with the lowest activation temperature ranging from 30 °C to 53 °C, while TRPV5-6 are highly Ca^2+^ selective channels that are not sensitive to temperature [7,8]. TRPV channels can sense various physical and chemical cues and are involved in multiple sensations, such as pain, itching, thermosensation, and mechanical stimuli [1]. TRPV1 is a founding member of the mammalian TRPV channel, and has been the most intensively studied. Extensive studies have revealed the role of TRPV1 in itching, pain, inflammation, and wound healing [9,10,11]. Recently, however, TRPV3 has received increasing attention because of its similar role to TRPV1.

TRPV3 is a non-selective cation channel that is highly expressed in keratinocytes in the skin. Traditionally, keratinocyte-expressed TRPV3 is involved in multiple physiological and pathological functions of the skin, such as itching, heat pain, and hair development. In addition, the mutation of the TRPV3 gene has been found to be responsible for the occurrence of diseases such as atopic dermatitis (AD) and Olmsted syndrome (OS). However, recent studies have suggested a novel role of TRPV3 in promoting skin regeneration, indicating that TRPV3 may become a new target for regulating skin regeneration.

This paper reviews the involvement of keratinocyte-expressed TRPV3 in the physiological and pathological processes, including itching, heat pain, hair development, human skin diseases, and skin regeneration.

## 2. The Characteristics of TRPV3 Channel

In humans, the TRPV3 gene is located at chromosome 17p13 near TRPV1, and the TRPV3 protein consists of 790 amino acids [3]. The human TRPV3 gene shows different degrees of sequence similarity to other TRPV family members, and the sequence similarity to TRPV1 and TRPV4 is the highest. The TRPV3 channel is a tetramer of the TRPV3 protein, composed of three domains: the *N*-terminal ankyrin repeat domain (ARD); the transmembrane domain, comprising six transmembrane helices (S1–S6), and the coupling domain, consisting of the TRP helix and the C-terminal β-strands [12]. TRPV3 is expressed in different human tissues, including the skin, brain, spinal cord, dorsal root ganglia (DRG), and testis [3]. In the skin, TRPV3 is highly expressed in keratinocytes from the basal layer of the epidermis and the outer root sheath (ORS) of the hair follicle [13,14,15,16,17]. Interestingly, TRPV3 was not detected in DRG neurons of mice, but was detected in those of humans [3,17].

The TRPV3 channel is a non-selective cation channel that can be activated by temperature (>33 °C) and a variety of plant-derived flavors, such as camphor, carvacrol, thymol, and eugenol [17,18,19]. TRPV3 channels have been found to mediate transport of Ca^2+^, Na^+^, K^+^, Cs+, and NH_4_^+^ [20,21]. The permeability of TRPV3 channels to Ca^2+^ is significantly stronger than that to other cations, and TRPV3-mediated I_Ca^2+^_ plays an important role in pain, itching, thermosensation, and inflammation [3,22,23,24]. The gating of TRPV3 is regulated by various molecules. For example, TRPV3 can be sensitized by repetitive stimulations of ligands such as 2-aminoethoxydiphenyl borate (2-APB), which is a commonly used activator of TRPV3, but can also activate TRPV1 and TRPV2 [25]. On the contrary, ATP reduces TRPV3 sensitivity, possibly through binding to the ARD domain of TRPV3 [26]. In addition, Mg^2+^ and phosphatidylinositol 4,5-bisphosphate (PIP2) were also found to inhibit TRPV3 sensitivity in primary epidermal keratinocytes [27,28].

## 3. Physiological and Pathological Functions of Keratinocytes-Expressed TRPV3

### 3.1. Itch

Itch is a common cutaneous sensation that is associated with an impaired skin barrier and inflammation [29,30]. Severe itching is a crucial feature of many cutaneous diseases and other systemic diseases, such as AD and OS, which severely diminish the quality of life of patients. Itch is transmitted by two pathways: histaminergic (in acute itch) and nonhistaminergic (in chronic itch) [31]. The histaminergic pathway depends on the release of histamines by (mainly) mast cells, basophils, and keratinocytes [32]. The non-histaminergic pathway is activated by pruritogens released by mast cells, granulocytes, macrophages, lymphocytes, keratinocytes, and neurons [32]. Keratinocytes participate in both pathways. Keratinocytes are also responsible for maintaining the integrity of the epidermal barrier to prevent exogenous pruritogens and epidermal inflammation [29]. G protein-coupled receptors and TRP channels are two classes of core receptors in the transmission of itching [32]. Numerous studies have shown that TRPV1 and TRPA1, from the TRP family, are two essential channels for itch sensation [31]. Recently, increasing studies have shown that keratinocytes-expressed TRPV3 is also involved in itch transmission (Figure 2) [33,34,35,36,37].

The first sign of TRPV3-mediated itching came from a mutant murine. DS-Nh mice and WBN/Kob-Ht rats are two kinds of autosomal dominant TRPV3 mutants, both of which appear as pruritus and other symptoms of AD [33,34]. Asakawa et al. [13] detected an amino acid substitution at the same position of the TRPV3 gene in both strains: Gly573 to Ser in Nh mice and Gly573 to Cys in Ht rats. Yoshioka et al. [35] created TRPV3 (Gly573Ser) transgenic mice and found that the transgenic mice showed spontaneous scratching behavior separately from the development of dermatitis, which indicates that the mutation of the TRPV3 gene is responsible for pruritus. Gly573Ser and Gly573Cys are gain-of-function mutations leading to increased TRPV3 channel activity in keratinocytes [38]. In addition to these evidence in murine models, more studies on humans further verified the role of TRPV3 in pruritus. In recent years, a number of clinical cases found that heterozygous missense mutations in the TRPV3 gene are the main cause of OS, which is characterized by severe itching, impaired hair growth, and keratoderma [39,40]. Furthermore, Yang et al. applied carvacrol, a TRPV3 activator, to human burn scars, which effectively elicited itching [37]. The enhanced expression of TRPV3 was also detected in the epidermis of burn scars with pruritus [36]. Collectively, these findings support the notion that TRPV3 in keratinocytes has a pivotal role in itch transmission (Table 1).

Although the underlying mechanisms of TRPV3 in chronic itch transmission remain unknown, recent reports have provided several clues (Figure 2). Seo et al. [41] found that the enhanced thermal sensitivity of TRPV3 in keratinocytes underlies heat-induced pruritogen release and pruritus in AD. They also found higher expression and enhanced activity of the TRPV3 channel in atopic lesional skin. TRPV3 channels in keratinocytes from AD lesional skin showed significantly higher sensitivity to innocuous heat (37 °C) compared with that from the healthy skin. After heat-induced TRPV3 activation, increased secretion of thymic stromal lymphopoietin (TSLP), nerve growth factor (NGF), and prostaglandin E2 receptor EP3 subtype-like (PGE2) by keratinocytes from AD lesional skin was observed [41]. Meanwhile, agonists of the TRPV3 channel also induced secretion of TSLP, NGF, PGE2, and IL33 in normal human keratinocytes and scratching behavior in mice [41]. These results indicate that TRPV3 is involved in warmth-provoked itching via these mediators.

Another study by Larkin et al. [42] showed that the B-type natriuretic peptide (BNP) is involved in the regulation of TRPV3 expression and activity in itch transmission. Previously, they verified that BNP is involved in itch transmission in AD [43]. In their latest study, they preincubated primary human keratinocytes with BNP, which resulted in increased TRPV3 expression and enhanced calcium response to the selective activator of TRPV3 [42]. Then, they found that BNP-provoked TRPV3 activation promoted the expression of Serpin E1. Upregulation of Serpin E1 in keratinocytes was also detected in lesional AD skin. Additionally, topical injection of Serpin E1 elicited scratching behavior in mice, and the Serpin E1 inhibitor attenuated scratching behavior in an AD-like chronic pruritus mouse model. All of these findings demonstrate that BNP-provoked TRPV3 activation participates in itch transmission through Serpin E1.

Inflammation may also play an essential role in TRPV3-mediated itch transmission [13,42,43]. In DS-Nh mice and WBN/Kob-Ht rats, all of which are gain-of-function TRPV3 mutation murine rodents, the skin showed an increased number of mast cells [13], indicating that inflammatory cytokines may also affect itch signaling. IL-31 can promote the synthesis and secretion of BNP, thus promoting itch transmission through BNP-provoked TRPV3 activation [42,43].

Although many important studies have been carried out, several questions remain to be resolved. From the bigger picture, how the pruritogens initiate itch transmission and activate TRPV3, as well as how itch signals are transmitted from keratinocyte-expressed TRPV3 to sensory neurons, remain unknown. Moreover, whether the differences in TRPV3 expression profiles in humans and mice indicates different TRPV3 mediated pruritus transmission processes deserves further investigations.

### 3.2. Heat Pain

As a member of TRPV subfamily similar to TRPV1, TRPV3 is suspected to have similar nociceptive sensation function to TRPV1, although this view is still controversial [15,18,44]. Most studies have supported the function of TRPV3 in nociceptive sensation. In human breast pain, increased TRPV1-positive nerve fibers and increased keratinocyte-expressed TRPV3 were found in the lesional epidermis [45]. In TRPV3 gain-of-function mutation disease, overactivated TRPV3 resulted in painful focal plantar keratoderma, which implies that TRPV3 may be involved in pain sensation, as TRPV1 is, but has different functions [44]. Morqrich et al. [15] knocked out TRPV1 and TRPV3 genes in mice, respectively, and found that TRPV1^−/−^ mice and TRPV3^−/−^ mice showed similarly delayed responses to acute noxious heat. In addition, the study also revealed that TRPV3 makes no contribution to pain provoked by mechanical stimuli or certain chemicals, such as formalin and bradykinin [15].

Some agonists and inhibitors have been found to have pain-regulating effects. Eugenol and carvacrol have been established to enhance noxious heat sensation, which that is described as a numbing and warm feeling with brief burning, stinging/pricking, and tingling [46]. Farnesyl pyrophosphate (FPP) has been found to be an endogenous TRPV3-selective agonist, and topical injection of FPP can elicit nociceptive behaviors in inflamed plantar tissue [47]. Isopentenyl pyrophosphate (IPP), a precursor molecule for FPP synthesis, inhibited TRPV3 and suppressed TRPV3-mediated acute nociception [48]. In addition, various local anesthetics (lidocaine, mepivacaine, ropivacaine, and bupivacaine) showed inhibitory effects on TRPV3 channels at pharmacologically relevant concentrations, suggesting that TRPV3 inhibition may be one of the mechanisms by which local anesthetics exert analgesic effects [49].

Research also showed that TRPV3-mediated nociception is relayed by the sensory neurons. Bang et al. [47] demonstrated that FPP activated mouse DRG neurons following the activation of keratinocytes in the co-culture system, but that FPP was unable to directly activate DRG neurons without keratinocytes. Since TRPV3 is also expressed in human DRG neurons, this finding only holds true for mice. Nevertheless, how TRPV3-activated keratinocytes transmit pain signals to neurons and whether there is a substance mediating the communications between keratinocytes and neurons is still worth investigation.

Although many authors agree that TRPV3 is involved in heat pain sensation, there has also been considerable research showing conflicting results. Huang et al. [50] studied the role of TRPV3 in heat pain sensation, but obtained opposite results to those of Moqrich et al. [15]; that is, TRPV3^−/−^ mice exhibited the same acute heat pain behavior as wild-type mice. They argue that the positive results from Moqrich et al. were caused by a mixed genetic background, and that TRPV3 makes a minimal contribution to heat pain sensation. Similar findings were also reported by Fatima et al. [51], who found that TRPV3 gain-of-function mutant mice exhibited the same response latency as wild-type mice to noxious heat. Furthermore, they evaluated other somatosensory behaviors and found desensitization to mechanical pain and cold. In summary, the role of TRPV3 in thermal nociception and other somatosensory processes needs to be further studied.

### 3.3. Hair Development

Abundant evidence has shown that keratinocyte-expressed TRPV3 is involved in natural hair development. TRPV3 gene deficiency and overactivation have different effects on hair, both of which impair hair development. TRPV3 knockout in mice led to hair irregularity (bent and curled) and keratinized cysts of hair follicles, suggesting that TRPV3 is responsible for maintaining normal hair morphology [15,52]. Meanwhile, in DS-Nh mice and WBN/Kob-Ht rats, over-activated TRPV3 resulted in a hairlessness phenotype, though a raised number of hair follicles were observed [13]. Altogether, these data demonstrate that appropriately-activated TRPV3 is the key to maintaining natural hair development.

Further studies revealed the regulatory effect of TRPV3 on hair follicles (Figure 3). TRPV3 was found to be expressed in multiple parts of hair follicles, including hair ORS, inner root sheath (IRS), and hair shaft [14,52,53]. Hair follicle keratinocytes proliferate, differentiate, and undergo apoptosis in a regular pattern to maintain normal morphogenesis, which, however, is affected by TRPV3 under pathological conditions [54,55]. TRPV3 overactivation suppresses proliferation and promotes apoptosis of hair follicle keratinocytes [14,52]. Using organ culture, Borbíró et al. [14] found that hair shaft elongation was significantly inhibited by TRPV3 activation of plant-derived agents, such as eugenol, thymol, and carvacrol. Similar results were reported by Song et al. [52], who found reduced proliferating cells in the hair matrix and an increase in the distal outer root sheath and infundibulum in TRPV3 knock-in mouse models. Additionally, TRPV3 activation also resulted in premature differentiation of hair follicle keratinocytes. Multiple keratins were diminished and multiple differentiation markers of the epidermis were significantly expanded in TRPV3 knock-in mice [52]. This impaired differentiation may relate to the increased apoptosis of hair keratinocytes induced by TRPV3 activation [14,53].

TRPV3 also participates in the regulation of the hair cycle [52,56]. Using in situ hybridization, results have shown that the expression of TRPV3 first appears in the differentiating cone of the hair follicles during early morphogenesis, peaks at the anagen, and becomes undetectable in the telogen [52]. In addition, the telogen observed 21 days after birth in normal DS mice was not observed in DS-NH mice, indicating that the prolonging of the anagen may also contribute to hair loss [56].

## 4. TRPV3 in Human Skin Diseases

Keratinocyte-expressed TRPV3 is associated with the occurrence of many skin diseases, such as OS, AD, psoriasis, post-burn pruritus, and rosacea, among which OS and AD are the most closely related to TRPV3 and have been studied the most extensively [16,24,57,58]. AD, also known as atopic eczema, is a recurrent, chronic inflammatory dermatosis characterized by persistent and intense pruritus. Many research studies have demonstrated that TRPV3 is involved in pruritus sensation in AD, which has been reviewed in the previous sections.

OS is a rare genodermatosis with a keratinizing disorder. Patients with OS often develop characteristic symptoms, such as extensive mutilating palmoplantar keratoderma, periorificial plaques, and alopecia, accompanied by severe pain and pruritus [59,60]. Other symptoms are also observed in some cases, such as onychodystrophy, oral leucokeratosis, corneal lesions, pseudoainhum, follicular keratosis, and erythromelalgia [59,61]. Mutations in the TRPV3 gene are known to be the main cause of OS, while mutations in the membrane-bound transcription factor peptidase, site 2 (MBTPS2) gene are responsible for a small number of cases [62]. Using a patch-clamp, Lin et al. [63] found that mutant TRPV3 channels were hyperactivated and mediated greater inward leakage currents. Zhong et al. [64] further compared the activity of different mutant types of TRPV3 channels and found that all of the mutants showed increased activity, but the increase of activity was different. The overactivation of TRPV3 channels arose out of gene mutation, resulting in a series of pathological changes in the skin of OS patients, including prominent hyperkeratosis, a thickened stratum spinosum layer with reduced stratum granulosum, disadhesion of cells in the suprabasal layers, elongation of rete ridges, and sparse lymphocyte infiltration in the dermis [65].

To date, various kinds of TRPV3 mutations have been identified (Table 2) [44,59,61,64,66]. Although all of these gain-of-function mutations lead to OS, different mutations manifest different levels of severity of symptoms [64]. For example, compared with general severe OS, the p.Gln580Pro mutation in TRPV3 only caused mild focal palmoplantar keratoderma (FPPK) [65]. This diversity in severity may be due to the different degrees of activation of the TRPV3 channel by different mutations. Similarly, the type of TRPV3 mutation also affects the inheritance of OS. In general, OS has mostly been described as sporadic, resulting from de novo mutations in TRPV3; indeed, in some families, OS has been found to be inherited in an autosomal dominant manner [63,67]. However, autosomal recessive inheritance has also been observed, in some cases, to be caused by a mutation of TRPV3. Eyton et al. [61] identified a homozygous recessive mutation, p.Trp521Ser, in TRPV3, which resulted in severe OS. Using protein modeling, they found that the recessive mutation p.Trp521ser affected the peripheral structure of TRPV3 more than the dominant mutation p.Gly573Ser; thus, it has less of an effect on TRPV3 function. This reminds us that the changes in TRPV3 channel structure resulting from amino acid substitutions also contribute to OS. Predicting the effect of gene mutations on TRPV3 activity through structural changes could accelerate the understanding of the etiology of OS.

In the past, the treatment of OS was very difficult. Often, doctors were only able to relieve the symptoms by surgically removing the cuticle and/or administering retinoids, steroids, urea, and body lotion (white petrolatum) [62]. However, these methods were only able to temporarily relieve some of the pain; the patient was still in distress. In 2020, Greco et al. [74] found the epidermal growth factor receptor (EGFR) inhibitor erlotinib to be a very effective drug for OS treatment. In their clinical trial, all patients who received erlotinib treatment had their hyperkeratosis and pain resolved and returned to normal life. This finding not only provides an effective treatment for patients, but suggests that EGFR signaling may be crucial for the pathogenesis of OS caused by the TRPV3 mutation.

## 5. Potential Target for Skin Regeneration

Previously, concerns about TRPV3 have focused on its role in itching, heat pain, and hair development. However, interestingly, accumulating evidence is revealing the important physiological function of TRPV3 in skin regeneration [65,75,76]. Many previous studies have demonstrated that TRPV3 significantly promotes cell proliferation. Zhang et al. [77] found that the increased expression and activation of TRPV3 promoted pulmonary artery smooth muscle cell proliferation in pulmonary arterial hypertension. Aijima et al. [75] found that TRPV3 activation elicited oral epithelia proliferation and promoted wound healing. More importantly, more research has revealed a strong promoting effect of TRPV3 on keratinocyte proliferation [65,76]. These results demonstrate the ability of TRPV3 to promote epidermal proliferation. Furthermore, TRPV3 has also been found to be involved in maintaining the skin barrier and angiogenesis [77,78]. Therefore, TRPV3 is a potential target for skin regeneration.

TRPV3 assists epidermal regeneration through EGFR signaling. EGFR, a tyrosine kinase, is involved in cell proliferation, division, and mitosis via initiating signaling cascade. EGFR can be activated by multiple ligands, such as amphiregulin (AREG), betacellulin (BTC), epidermal growth factor (EGF), heparin-binding EGF-like growth factor (HB-EGF), and transforming growth factor-α (TGF-α) [79]. It is well established that EGFR promotes wound healing and skin regeneration via promoting the proliferation and migration of epidermal cells [80,81,82,83]. Recent studies have revealed the interplay between TRPV3 and EGFR in skin regeneration (Figure 4). Aijma and Wang et al. [75,76] found that TRPV3 activation promoted EGFR phosphorylation, and that the inhibition of EGFR suppressed the proliferation of HaCaT cells, suggesting that TRPV3 promoted epidermal proliferation by activating EGFR. At the same time, activated EGFR signaling in turn activated TRPV3 via ERK in a positive feedback manner [84]. Cheng et al. [78] found that TRPV3 interacted with ligands of EGFR. On the one hand, the activation of TRPV3 by agonists elicited a greater release of TGF-α in human keratinocytes. On the other hand, TGF-α/EGF treatment enhanced the sensitivity of TRPV3 to agonists through the EGFR-PLC/ERK pathway in keratinocytes. Utilizing coimmunoprecipitation experiments, they also confirmed that TRPV3 and EGFR formed a signaling complex in which TGF-α elicited EGFR activation, resulting in the tyrosine phosphorylation of TRPV3. In fact, it is the Ca^2+^ influx after TRPV3 activation that played a practical role in these processes, by which increased Ca^2+^ activated CaMKII, which subsequently induced EGFR activation and TGF-α release [76]. Phosphatidylinositol 4,5-biphosphate (PIP2) is hydrolyzed into IP3 and diacyl glycerol (DAG) by phospholipase C (PLC). Doerner et al. [28] found that PLC could improve TRPV3 sensitivity by hydrolyzing PIP2. This study complements the finding by Cheng et al. that agonists activate TRPV3 through the EGFR-PLC-PIP2 pathway [28,78]. In addition to EGFR signaling, another factor affecting TRPV3 sensitivity is cholesterol. Klein et al. [85] found that cholesterol enrichment could improve the sensitivity of TRPV3, which then facilitates TRPV3 activation by a lower concentration of activators or a lower temperature. This regulatory mechanism may relate to the activation of EGFR and PLC by cholesterol enrichment.

The phosphatidylinositol 3 kinase (PI3K)/protein kinase B (AKT) pathway has been recognized as the downstream signaling pathway of EGFR activation induced by TRPV3 [77]. The PI3K/AKT signaling pathway has been suggested to promote skin homeostasis and skin regeneration by promoting cell proliferation and epithelial–mesenchymal transition (EMT), as well as by inhibiting apoptosis [86,87]. The inhibition of PI3K/AKT signaling suppressed epidermal hyperplasia induced by TRPV3 agonists [76,77]. In addition, nuclear factor kappa-B (NF-κB) was also activated by EGFR, which was activated by TRPV3 [76]. Though some previous studies suggested that NF-κB was activated by PI3K/AKT signaling, the dependence of NF-κB activation induced by TRPV3 on PI3K/AKT needs to be studied further [88,89].

TRPV3 may accelerate skin regeneration by regulating nitric oxide (NO) synthesis. Plenty of evidence has demonstrated the positive role of NO in wound healing. An appropriate level of NO can promote skin regeneration by modulating cell proliferation, differentiation, and apoptosis [90]. NO also works in the inflammatory phase, angiogenesis, and tissue remodeling process, thus accelerating wound healing [90]. It has been found that low levels of NO can promote keratinocyte proliferation, while high levels of NO suppress cell proliferation in the epidermis [91]. In addition, NO can also regulate keratinocyte differentiation and inhibit cell apoptosis [92]. Normally, NO can be synthesized from L-arginine catalyzed by nitric oxide synthase [92]. However, a recent study also showed that the activation of TRPV3 can induce NO production in a NOS-independent way [93]. Miyamoto et al. [93] found that neither NOS inhibitors nor NOS gene deficiency have an effect on NO production induced by TRPV3 activation, although NO production is dependent on the presence of nitrite. Collectively, these results indicate that TRPV3 promotes skin regeneration by inducing NO production in a nitrite-dependent way. However, more investigations are needed.

As a new therapeutic target for skin regeneration, TRPV3 has many advantages. For example, TRPV3 is a promising target for improving wound healing. In the epidermis, appropriately-activated TRPV3 can promote keratinocyte proliferation, differentiation, and migration and inhibit cell apoptosis, thus accelerating re-epithelialization [76,94]. In the dermis, TRPV3 is involved in the process of dermal extracellular matrix deposition, which can improve the integrity of injured skin, which also suggests the potential of TRPV3 in regulating scar formation [95]. Meanwhile, TRPV3 can also improve wound healing by promoting angiogenesis and vascular remodeling [77,93].

Nevertheless, some problems interfere with the application of TRPV3 as a therapeutic target. Overactivation of TRPV3 is related to itch sensation and various skin diseases, such as AD and OS. Therefore, finding the right dose of therapeutic drugs (in this case, TRPV3 activators) to promote skin regeneration while minimizing the side effects can be a challenge.

## 6. Conclusions

This article reviews the roles of keratinocyte-expressed TRPV3 in itching, heat pain, and hair development in both rodent and human tissues. Although solid evidence has been obtained from TRPV3 mutant and transgenic mice, the poor selectivity of TRPV3 modulators did undermine reliability. Thus, for future studies, TRPV3 modulators with a higher level of selectivity are needed. This article also summarizes the mechanisms by which TRPV3 functions in these physiological and pathological processes associated with TRPV3 and highlights EGFR signaling. In addition, the potential role of TRPV3 in promoting skin regeneration has also been emphasized. Although TRPV3 modulators may serve as a promising therapeutic medication for related skin diseases, its limitations, such as its side effects, cannot be ignored. In conclusion, this review lays a foundation for further developing our understanding of the mechanisms by which TRPV3 is involved in itching, heat pain, and hair development, as well as the treatments for TRPV3-related skin diseases.

## Figures and Tables

**Figure 1 biomolecules-13-00513-f001:**
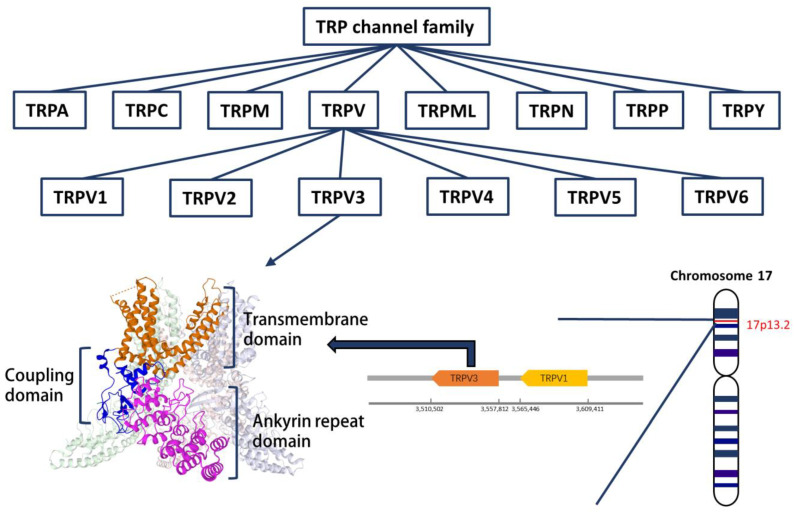
TRP channel family and TRPV3 channel protein. TRP channel, transient receptor potential cation channel. TRPA, TRP channel subfamily A (ankyrin). TRPC, TRP channel subfamily C (canonical). TRPM, TRP channel subfamily M (melastatin). TRPML, TRP channel (mucolipin). TRPN, TRP channel subfamily N (Drosophila NOMPC). TRPP, TRP channel subfamily P (polycystin). TRPV, TRP channel subfamily V (vanilloid). TRPY, TRP channel subfamily Y (yeast). (TRPV3 structure image is cited from Shimada, H., Kusakizako, T., Nishizawa, T., Nureki, O. (2020) cryo-EM structure of TRPV3 in lipid nanodisc. https://doi.org/10.2210/pdb6LGP/pdb accessed on 02 February 2023).

**Figure 2 biomolecules-13-00513-f002:**
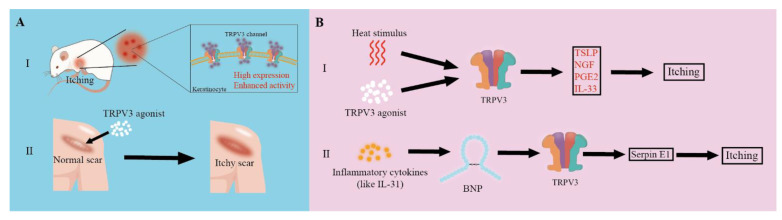
(**A**) Evidence of TRPV3-mediated itching. (**I**) High expression and enhanced activity of the TRPV3 channel was found in itchy skin. (**II**) After treatment with TRPV3 agonist, the normal scar became itchy. (**B**) The mechanism of TRPV3-mediated itching. (**I**) After activation by heat stimulus or a TRPV3 agonist, activated TRPV3 provoked itching through thymic stromal lymphopoietin (TSLP), nerve growth factor (NGF), prostaglandin E2 receptor EP3 subtype-like (PGE2), and IL-33. (**II**) Inflammatory factors such as IL-31 induce high expression of B-type natriuretic peptide (BNP) to activate TRPV3, which leads to the upregulation of Serpin E1 and itching.

**Figure 3 biomolecules-13-00513-f003:**
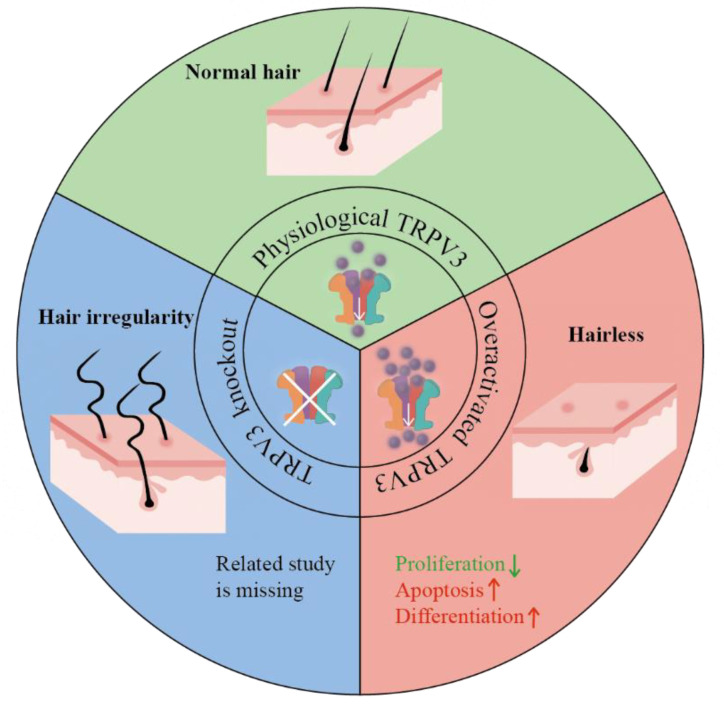
The regulatory effect of TRPV3 in hair development. Physiological TRPV3 maintains normal hair growth. Overactivated TRPV3 results in decreased proliferation, increased apoptosis, and increased differentiation of cells, as well as the hairless phenotype. TRPV3 knockout leads to hair irregularity.

**Figure 4 biomolecules-13-00513-f004:**
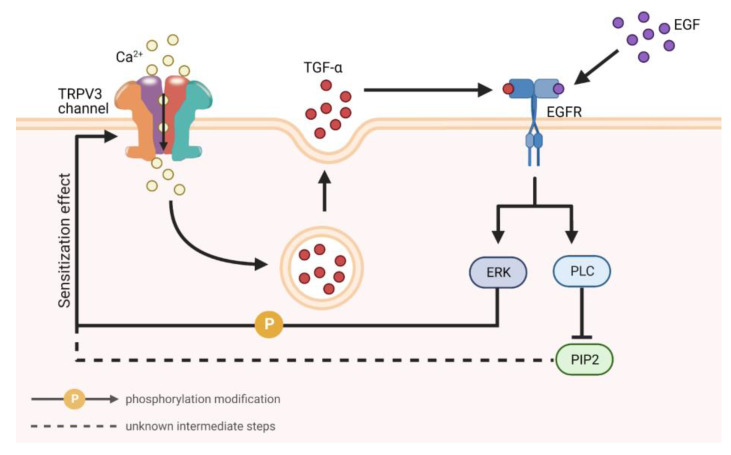
Positive feedback activation of TRPV3 and EGFR. The opening of activated TRPV3 channels for Ca^2+^ influx leads to a surge in intracellular calcium signaling, which promotes TGF-α release. TGF-α and EGF in turn activate their receptor EGFR, which then activates ERK and PLC. ERK sensitizes TRPV3 by phosphorylation modification and PLC by hydrolyzing PIP2. TRPV3 in the sensitized state is more easily activated by stimuli. In a word, TRPV3 and EGFR form a positive feedback activation pathway.

**Table 1 biomolecules-13-00513-t001:** Evidence for TRPV3-mediated itching.

Authors	Year	Results	Summary
Asakawa et al. [33]	2005	WBN/Kob-Ht rats spontaneously developed dermatitis under conventional conditions	In rodents, mutations in genes caused increased TRPV3 activity, leading to pruritus and dermatitis
Watanabe et al. [34]	2003	DS-Nh mice spontaneously developed dermatitis under conventional conditions	
Asakawa et al. [13]	2006	TRPV3 (Gly573Ser) mutation was detected in Nh mice, and TRPV3 (Gly573Cys) mutation in Ht rats	
Yoshioka et al. [35]	2009	Artificially constructed TRPV3 (Gly573Ser) transgenic mice showed spontaneous scratching behavior, separately from the development of dermatitis	
Xiao et al. [38]	2008	Gly573Ser and Gly573Cys are gain-of-function mutations leading to increased TRPV3 channel activity	
Yang et al. [37]	2015	Application of TRPV3 activator carvacrol to burn scars caused itching	In humans, activation of TRPV3 causes itching in burn scars
Kim et al. [36]	2020	Enhanced expression of TRPV3 was detected in the epidermis of burn scars with pruritus	

**Table 2 biomolecules-13-00513-t002:** TRPV3 mutations that cause Olmsted syndrome.

Mutation	Mode of Inheritance	Severity	Reference
p.Asn415_Arg416insLeuAsn	Autosomal dominant	Severe	Agarwala et al. [59], 2016
p.Arg416Gln	Autosomal dominant	Moderate	Zhong et al. [64], 2021
p.Arg416Trp	Autosomal dominant	Moderate	Zhong et al. [64], 2021
p.Trp521Ser	Autosomal recessive	Severe	Eytan et al. [61], 2014
p.Gly568Cys	Autosomal recessive	Severe	Duchatelet et al. [60], 2014
p.Gly568Asp	Autosomal dominant	Moderate	Peters et al. [44], 2020
p.Gly568Cys/p.Gly215Vfs*82	Autosomal semidominant	Severe	Cao et al. [68], 2016
p.Gly568Val	Autosomal dominant	Severe	Nagai et al. [66], 2017
p.Gly573Ser	Autosomal dominant	Severe	Lai-Cheong et al. [69], 2012 and Lin et al. [63], 2012
p.Gly573Cys	Autosomal dominant	Moderate	Lin et al. [63], 2012
p.Gly573Ala	Autosomal dominant	Severe	Danso-Abeam et al. [70], 2013
p.Gly573Val	Autosomal dominant	Severe	Zhi et al. [71], 2016
p.Gln580Pro	Autosomal dominant	Moderate	He et al. [65], 2015
p.Leu655Pro	Autosomal dominant	Moderate	Zhong et al. [64], 2021
p.Met672IIe	Autosomal dominant	Severe	Ni et al. [67], 2016
p.Leu673Phe	Autosomal dominant	Moderate	Duchatelet et al. [72], 2014
p.Ala675Thr	Autosomal dominant	Severe	Chiu et al. [39], 2020
p.Trp692Gly	Autosomal dominant	Moderate	Lin et al. [63], 2012
p.Trp692Cys	Autosomal dominant	Severe	Kariminejad et al. [73], 2014
p.Trp692Ser	Autosomal dominant	Moderate	Zhong et al. [64], 2021
p.Leu694Pro	Autosomal dominant	Mild	Zhong et al. [64], 2021

## Data Availability

Not applicable.

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
