# Peer review of "Novel Insights into the Role of Keratinocytes-Expressed TRPV3 in the Skin"

_biomolecules, 2023, doi:10.3390/biom13030513_

Round 1

Reviewer 1 Report

The authors provide a detailed description of the role of TRPV3 in skin conditions/health. The manuscript is well written but would benefit from major revisions. Suggested edits are included below.

Major revisions:

For Figures 1-3: How were these figures generated? Was it generated using software or was it adapted from other publications? If software was used, this should be listed in the acknowledgement section or if the images are adaptations of published images the sources/and permissions should be included. The acknowledgement section only mentions the use of Biorender for Figure4. Also, in this reference to the use of Biorender for figure 4, the permit or license number should be provided (or submitted along with the manuscript).

Figure 2: The color boxes are each labeled with a capital A and B. However, the individual part sof the figure are again labeled as a. and b. This is confusing. I would perhaps label the subfigures with numbers or a different system. This is particularly confusing in the legend. B. a has a description, but B.b. does not.

Figure 3: Does a good job summarizing findings as they relate to the different hair conditions.

Line 106-107: The authors mention that there are increasing studies finding that keratinocytes expressed TRPV3 is also involved in itch. They even include Figure 2 to depict this. However, no studies are cited. Please add corresponding citations. The next paragraph includes citations, but it is unclear if this sentence is referring to the next paragraph. If so, please include citations in this sentence as well.

Lines 114-115: This paragraph lists studies examining TRPV3’s role in itch. I would suggest that these studies are summarized in a table of evidence that includes: The study (authors, year) and a summary of the finding from each study and how they fit together.

Lines 297-298: If the role of TRPV3 is already well established, the importance of this review is diminished. If emerging literature has contributed to a better understanding of the role of TRPV3, then it is not fully established. This is contradictory- is it well accepted or is emerging literature expanding our understanding of the role of this protein?

Lines 369-371: This sentence states that TRPV3 is the cause of AD and OS. However, there are other pathophysiological conditions that cause AD and OS. These diseases may be characterized by signs and symptoms that involve TRPV3, but they are not caused by this receptor.

Lines 370-371: This sentence is missing a noun. What is “the right dose of”? The right dose of a treatment that involves TRPV3 (e.g., a TRPV3 agonist perhaps)?  Also, there are very few sections of the review that talk about treatment. It is unclear to me why this is highlighted at the end of the discussion. Perhaps a more in-depth discussion on treatments should be included in preceding sections.

Lines 373-378: The sentences in the conclusion should be rephrased. They are difficult to understand. Perhaps “the mechanism by which TRPV3 promotes…..”. The conclusion is also very vague and needs further detail to sufficiently capture the overall findings of the review. More details are needed to explain how all the data described and presented builds upon our understanding of the role of TRPV3 on skin conditions. Additionally, it would be important to discuss the clinical implications of the advancements in our understanding of the role of TRPV3. This section should summarize how this review advances or recapitulates our knowledge about TRPV3 and why TRPV3 is important to health. The last sentence describes that “This review will lay a foundation for further understanding the mechanisms by which TRPV3 is involved in itch, heat pain, and hair development, as well as the treatments for TRPV3-related skin diseases.” This would be a good place to summarize how this review is laying the foundation or our understanding of TRPV3-related skin diseases.

I would recommend that the manuscript is reviewed for English language and style. It is for the most part, well written. However, there are multiple instances where the language should be revised to be more formal or it is incorrect. Some of these instances are included below (in minor revision section).

General:

-The authors often refer to different types of epithelia (e.g., oral). How is the role of TRPV3 different in different tissue types?

-There are multiple occasions where the authors include only 1 citation to support a finding. However, often it requires multiple studies to establish a particular role or function. There needs to be discussion acknowledging that there may be insufficient evidence to support a conclusion or more studies need to provided.

Minor Revisions:

Line 18: “researches” is incorrectly used. Are the authors referring to “research studies”?

Line 19: “increasing literatures” is incorrect. Literature should not have an “s”. Are authors referring to “an increasing number of studies” ? This also applies to line 63 and throughout the manuscript. Literature is plural, unless referring to different types of literature. However, in this manuscript authors seem to be referring only to scientific literature as whole.

Line 47: What is a “mammalian member”? This is the incorrect term. Are the authors referring to transmembrane proteins?

Line 96: “living quality” should be “quality of life”

Lines 164-165: The sentence ends in a question. Please rephrase. If this is indeed a question that remains to be answered please rephrase as such. This also applies to sentences 192-195.

Line 205: I am unsure as to what is meant by “in a word”. Is this referring to “in Summary”?

Line 208-209: This sentence is written very informally. Please remove the word “lots” and replace with another more descriptive and appropriate term.

Line 267-268: “up to date” is incorrectly used. This sentence is also missing citations. Specially if referring to multiple sources.

Lines 307-308: This sentence should be rephrased. The authors should clarify whether they are basing this inference on the data or speculation (believe). If this is an inference made from the data described in the previous sentence, then they should remove “we believe and come up with that...”. It is also not grammatically correct.  

Line 361-362: Incomplete sentence. Comma may be added after TRPV3.

Line 369: Sentence needs to be rephrased. It is unclear what is meant by “the application of TRPV3”. Is it the application of a treatment that involves TRPV3?

Reviewer 2 Report

This condensed review focuses on physiological and pathophysiological functions of TRPV3 in the skin - mainly by contributing to Ca2+ flux in keratinocytes. Although genetic evidence from mutated TRPV3 channels that exert gain-of- or loss-of-function variants clearly points to a seminal role of TRPV3 in skin homeostasis, the physiological regulatory mechanisms are only partially understood, and most pharmacological modulators are poorly selective, thus causing misleading conclusions.

The language needs to be corrected in many places, e.g. singular/plural inconsistencies, lacking articles or pronouns.

Among the few mechanisms by which TRPV3 activity is modulated by physiological mechanisms, the sensitisation of TRPV3 by cholesterol should be added, and the possible link between PLC-dependent channel sensitisation and PIP2-dependent inhibition may be discussed.

The table containing all single Olmsted mutants is of limited value. Instead, a table referring to the specific TRPV3-linked diseases, including proposed mechanisms by which TRPV3 is involved, may be added.

There are a few inconsistencies and corrections that may require correction:

line 29: most TRP channels are not Ca2+(-selective) channels.

line 31: "about 30 TRP proteins" and "100 TRP channels": please specify these numbers for mammals (which have keratinocytes) and omit drosophila or yeast TRPs

lines 34 and 71: homology is a qualitative term. You may wish to replace the term "homology" by "sequence similarity".

line 69/70: genes or chromosomes do not consist of amino acids

figure 2: correct "Seprin E1"

line 223: correct the writing of the name Borbiro

Figure 3 and line 300: there is an incensistency of lower proliferation in (oder)activated TRPV3 and promotion of cell proliferation via TRPV3 activation.

Figure 4: PLC is not a protein kinase. Thus, the arrow from PLC via (P) towards TRPV3 may be misleading. Interrupted arrows may indicate the as yet unknown intermediate steps.

line 335: typo "EFGR"

references: inconsistent use of uppercase annotations in the titles of cited studies.
